# Correlator convolutional neural networks as an interpretable architecture for image-like quantum matter data

Cole Miles [1], Annabelle Bohrdt [2,3,4], Ruihan Wu[5], Christie Chiu [2,6,7], Muqing Xu[2], Geoffrey Ji [2], Markus Greiner[2], Kilian Q. Weinberger[5], Eugene Demler[2] & Eun-Ah Kim [1✉]

Image-like data from quantum systems promises to offer greater insight into the physics of correlated quantum matter. However, the traditional framework of condensed matter physics lacks principled approaches for analyzing such data. Machine learning models are a powerful theoretical tool for analyzing image-like data including many-body snapshots from quantum simulators. Recently, they have successfully distinguished between simulated snapshots that are indistinguishable from one and two point correlation functions. Thus far, the complexity of these models has inhibited new physical insights from such approaches. Here, we develop a set of nonlinearities for use in a neural network architecture that discovers features in the data which are directly interpretable in terms of physical observables. Applied to simulated snapshots produced by two candidate theories approximating the doped Fermi-Hubbard model, we uncover that the key distinguishing features are fourth-order spin-charge correlators. Our approach lends itself well to the construction of simple, versatile, end-to-end interpretable architectures, thus paving the way for new physical insights from machine learning studies of experimental and numerical data.

[1] Department of Physics, Cornell University, Ithaca, NY, USA. [2] Department of Physics, Harvard University, Cambridge, MA, USA. [3] Department of Physics and Institute for Advanced Study, Technical University of Munich, Garching, Germany. [4] Munich Center for Quantum Science and Technology (MCQST), München, Germany. [5] Department of Computer Science, Cornell University, Ithaca, NY, USA. [6] Department of Electrical Engineering, Princeton University, Princeton, NJ, USA. [7] Princeton Center for Complex Materials, Princeton University, Princeton, NJ, USA. ✉email: eun-ah.kim@cornell.edu

There have been growing efforts to adopt data science tools that have proved effective at recognizing every-day objects for objective analysis of image-like data on quantum matter[1–6]. The key idea is to use the ability of neural networks to express and model functions to learn key features found in the image-like data in an objective manner. However, there are two central challenges to this approach. First, the "black box" nature of neural networks is particularly problematic when it comes to scientific applications, where it is critical that the outcome of the analysis is based on scientifically correct reasoning[7]. The second challenge unique to scientific application of supervised machine learning (ML) approaches is the shortage of real training data. Hence the community has generally relied on simulated data for training[1,3,8]. However, it has not been clear whether the neural networks trained on simulated data properly generalize to experimental data. The path to surmounting both of these issues is to obtain some form of interpretability in our models. To date, most efforts at interpretable ML on scientific data have relied on manual inspection and translation of learned features from training standard architectures[9–11]. Instead, here we propose an approach designed from the ground-up to automatically learn information that is meaningful within the framework of physics.

The need for a principled data-centric approach is particularly great and urgent in the case of synthetic matter experiments such as quantum gas microscopy (QGM)[12–14], ion traps[15], and Rydberg atom arrays[16,17]. While our technique is generally applicable, in this work we focus on QGM, which enables researchers to directly sample from the many-body density matrix of strongly correlated quantum states that are simulated using ultra-cold atoms. With the quantum simulation of the fermionic Hubbard model finally reaching magnetism[14] and the strange metal regime[18,19], QGM is poised to capture a wealth of information on this famous model that bears many open questions and is closely linked to quantum materials. However, the real-space snapshots QGM measures are a fundamentally new form of data resulting from a direct projective measurement of a many-body density matrix as opposed to a thermal expectation value of observables. While this means richer information is present in a full dataset, little is known about how to efficiently extract all the information. When it comes to the questions regarding the enigmatic under-doped region of the fermionic Hubbard model, the challenge is magnified by the fact that fundamentally different theories can predict QGM data with seemingly subtle differences within standard approaches[19,20].

In this work, we develop Correlator Convolutional Neural Networks (CCNNs) as an architecture with a set of nonlinearities which produce features that are directly interpretable in terms of correlation functions in image-like data (see Fig. 1). Following training of this architecture, we employ regularization path analysis[21] to rigorously identify the features that are critical in the CCNN's performance. We apply this powerful combination of CCNNs and regularization path analysis to simulated QGM data of the under-doped Fermi-Hubbard model, as well as additional pedagogical examples in Supplementary Note 2. Following this, we discuss the new insights we gain regarding the hidden signatures of two theories, geometric string theory[22] and $\pi$-flux theory[23,24].

## Results

The Hubbard model of fermionic particles on a lattice is a famous model that bears many open questions and is closely linked to quantum materials such as high-temperature superconductors. The model Hamiltonian is given by

$$\mathcal{H} = -t \sum_{\sigma=\uparrow,\downarrow} \sum_{\langle i,j \rangle} (\hat{c}_{i,\sigma}^{\dagger} \hat{c}_{j,\sigma} + \text{h.c.}) + U \sum_{i} \hat{n}_{i,\uparrow} \hat{n}_{i,\downarrow} \quad (1)$$

where the first term describes the kinetic energy associated to electrons hopping between lattice sites, and the second term describes an on-site repulsion between electrons. At half-filling, and in the limit $U \gg t$, the repulsive Hubbard model maps to the Heisenberg antiferromagnet (AFM)[25]. However, the behavior of

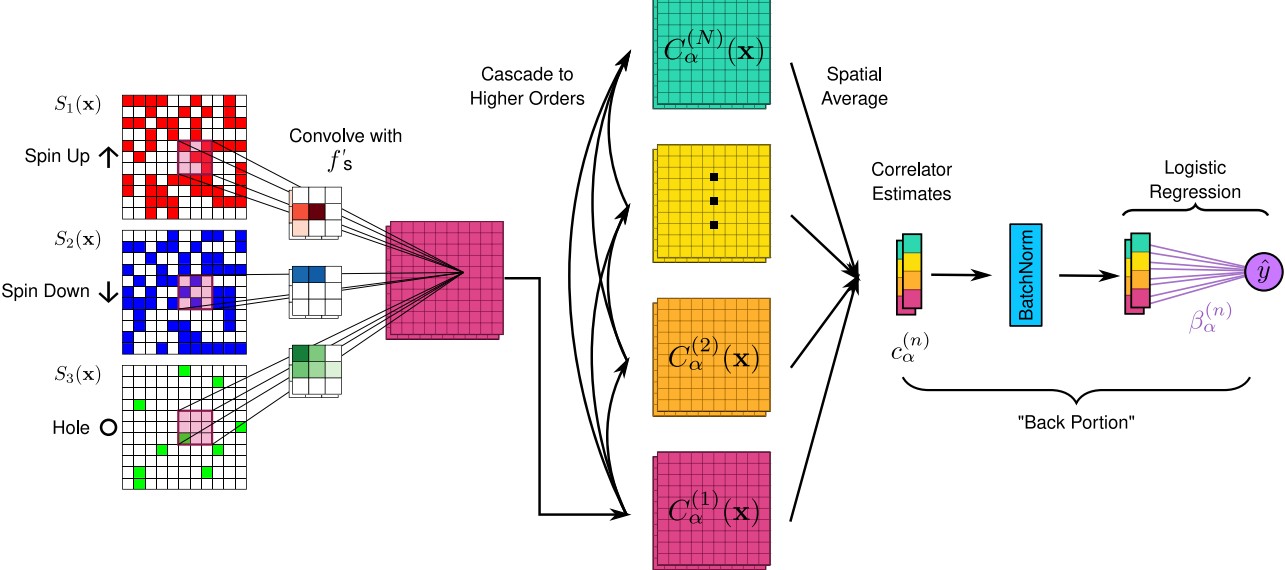

**Fig. 1 Correlator Convolutional Neural Network Architecture.** The construction of our Correlator Convolutional Neural Network, shown here with two learnable filters ($M = 2$). The input is a three-channel image: $S_1(\mathbf{x}) = n_{\uparrow}(\mathbf{x})$, $S_2(\mathbf{x}) = n_{\downarrow}(\mathbf{x})$, $S_3(\mathbf{x}) = n_{\text{hole}}(\mathbf{x}) = 1 - S_1(\mathbf{x}) - S_2(\mathbf{x})$. Note that $S_3$ is redundant information, but is provided for improved performance and interpretability. The image is first convolved with learned filters $f_\alpha$ to produce a set of convolutional maps $C_\alpha^{(1)}(\mathbf{x})$. Maps containing information about higher-order local correlations can then be recursively constructed using the lower-order maps, truncating at some order $N$. Spatially averaging these maps produces features $c_\alpha^{(n)}$ which in expectation are equal to weighted sums of correlators found as subpatterns of the corresponding convolutional filter. These features are normalized to zero mean and unit variance by a BatchNorm layer, then used by a logistic classifier with coefficients $\beta_\alpha^{(n)}$ to produce the final output $\hat{y}$.

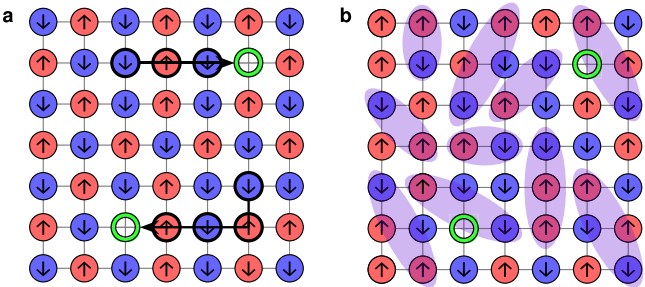

**Fig. 2 Model effective theories of the 2D Fermi-Hubbard model.** A cartoon depicting the features of two candidate theories approximating the low-$T$, low-doping limit of the Fermi-Hubbard model. Red, blue sites are spin-up, spin-down electrons respectively, while green circles represent holes. **a** Geometric string theory, showing two geometric strings in the presence of an antiferromagnetic background. Note that the propagation of the doped holes creates parallel line segments of aligned spins, perpendicular to the direction of the hole propagation. **b** $\pi$-flux theory, which describes a spin liquid of singlet pairs.

the model as the system is doped away from half-filling is not as well-understood. Several candidate theories exist which attempt to describe this regime, including geometric string theory[22] and $\pi$-flux theory[23,24]. These theories are conceptually very distinct, but at low dopings measurements in the occupation basis do not differ enough in simple conventional observables such as staggered magnetization or two-point correlation functions to fully explain previous ML success[3] in discrimination (see Supplementary Note 4). Nevertheless, there are more subtle hidden structures involving more than two sites[20] which are noticeable. In the "frozen spin approximation"[26], geometric string theory predicts that the motion of the holes simply displaces spins backwards along the path the hole takes. Hence the propagation of the doped hole will tend to produce a "wake" of parallel line segments of aligned spins in its trail (Fig. 2(a)). Meanwhile, the $\pi$-flux theory describes a spin liquid of singlet pairs, where it is more difficult to conceive of characteristic structures (Fig. 2(b)).

Current QGM experiments are able to directly simulate the Fermi-Hubbard model, obtaining one or two-dimensional occupation snapshots sampled from the thermal density matrix $\rho \sim e^{-\beta \mathcal{H}}$ prescribed by the model[14]. However, currently our experiment can only resolve a single spin species at a time, leaving all other sites appearing as empty. This is not a fundamental limitation of QGM experiments and complete spin and charge readout is beginning to become available to select groups[27,28]. As we aim to learn true spin correlations, in this work we use primarily simulated snapshots at doping $\delta = 0.09$ sampled from the geometric string and $\pi$-flux theories using Monte-Carlo sampling techniques under periodic boundary conditions. In particular, geometric string snapshots are generated by first sampling snapshots from the AFM Heisenberg model, then randomly inserting strings with lengths drawn from the analytic distribution[20]. $\pi$-flux snapshots are generated by standard Metropolis sampling from the Gutzwiller projected thermal density matrix given by the associated mean-field Hamiltonian. (See Supplementary Note 1 for further details.)

We point out that in the context of this paper, when referring to two models as different, we do not imply that they are fundamentally distinct, in the sense that they can not be connected smoothly without encountering a singularity in the partition function. Rather, this is a practical question: we have two or more mathematical procedures for generating many-body snapshots based on variational wavefunctions, Monte-Carlo sampling, or any other theoretical approach. Our goal is to develop a ML

algorithm that separates snapshots based on which procedure they are more likely to come from and, most importantly, the algorithm should provide information about which correlation functions are most important for making these assignments.

To learn how to distinguish these two theories we propose a neural network architecture, CCNNs, schematically shown in Fig. 1. The input to the network is an image-like map with 3-channels $\{S_k(\mathbf{x})|k = 1, 2, 3\}$, where $S_1(\mathbf{x}) = n_\uparrow(\mathbf{x})$, $S_2(\mathbf{x}) = n_\downarrow(\mathbf{x})$, $S_3(\mathbf{x}) = n_{\text{hole}}(\mathbf{x})$. Since the models we consider are restricted to the singly-occupied Hilbert space, this input only takes on values 0 or 1. From this input, the CCNN constructs nonlinear "correlation maps" containing information of local spin-hole correlations up to some order $N$ across the snapshot. This operation is parameterized by a set of learnable 3-channel filters, $\{f_{\alpha,k}|\alpha = 1, \cdots, M\}$ where $M$ is the number of filters in the model. The maps for the given filter $\alpha$ are defined as:

$$
\begin{aligned}
C_\alpha^{(1)}(\mathbf{x}) &= \sum_{\mathbf{a},k} f_{\alpha,k}(\mathbf{a})S_k(\mathbf{x} + \mathbf{a}) \\
C_\alpha^{(2)}(\mathbf{x}) &= \sum_{(\mathbf{a},k)\neq(\mathbf{b},k')} f_{\alpha,k}(\mathbf{a})f_{\alpha,k'}(\mathbf{b})S_k(\mathbf{x} + \mathbf{a})S_{k'}(\mathbf{x} + \mathbf{b}) \\
&\vdots \\
C_\alpha^{(N)}(\mathbf{x}) &= \sum_{(\mathbf{a}_1,k_1)\neq\cdots\neq(\mathbf{a}_N,k_N)} \prod_{j=1}^N f_{\alpha,k_j}(\mathbf{a}_j)S_{k_j}(\mathbf{x} + \mathbf{a}_j).
\end{aligned}
\tag{2}
$$

Here $\mathbf{a}$ runs over the convolutional window of the filter $\alpha$. Traditional convolutional neural networks employ only the first of these operations, alternating with some nonlinear activation function such as tanh or $\text{ReLU}(x) = \max(0, x)$. The issue with these typical nonlinear functions is that they mix all orders of correlations into the output features, making it difficult to disentangle what exactly traditional networks measure. In contrast, each order of our nonlinear convolutions $C_\alpha^{(n)}(\mathbf{x})$ are specifically designed to learn $n$-site semi-local correlations in the vicinity of the site $\mathbf{x}$, which appear as patterns in the convolutional filters $f_\alpha$. Note that the sums in Eq. (2) exclude any self-correlations to aid interpretability. During training, a CCNN tunes the filters $f_{\alpha,k}(\mathbf{a})$ such that correlators characteristic of the labeled theory are amplified while others are suppressed. To aid interpretation, we force all filters to be positive $f_{\alpha,k}(\mathbf{a}) \geq 0$ by taking the absolute value before use on each forward pass. We note that a multi-site kernel used in a support vector machine, as introduced in refs. [29,30], could also learn higher-order correlators. However, CCNNs allow high-order correlations to be efficiently parameterized and discovered by leveraging automatic differentiation and the structure of convolutions.

A direct computation of the nonlinear convolutions following Eq. (2) up to order $N$ requires $\mathcal{O}((KP)^N)$ operations per site, where $P$ is the number of pixels in the window of the filter and $K$ is the number of species of particles. However, we can use the following recursive formula which we prove in Supplementary Note 3:

$$
C_\alpha^{(n)}(\mathbf{x}) = \frac{1}{n}\sum_{l=1}^n (-1)^{l-1} \left(\sum_{\mathbf{a},k} f_{\alpha,k}(\mathbf{a})^l S_k(\mathbf{x} + \mathbf{a})^l\right) C_\alpha^{(n-l)}(\mathbf{x})
\tag{3}
$$

where all powers are done pixelwise, and we define $C_\alpha^{(0)}(\mathbf{x}) = 1$. This improves the computational complexity to $\mathcal{O}(N^2KP)$ while also allowing us to leverage existing highly-optimized GPU convolution implementations. Use of this formula leads to a "cascading" structure to our model similar to[31], as seen in Fig. 1. First, the input $S$ is convolved with filters $f_\alpha$ to produce the first-order maps $C_\alpha^{(1)}$. Using Eq. (3), these first-order maps can be used to construct second order maps $C_\alpha^{(2)}$, and onwards until the model is truncated at some order $N$. Since the Hamiltonians being studied are translation-invariant, we then obtain estimates of

correlators from these correlation maps by simple spatial averages to produce $c_\alpha^{(n)} = \frac{1}{N_{\text{sites}}} \sum_{\mathbf{x}} C_\alpha^{(n)}(\mathbf{x})$. In addition, we employ an explicit symmetrization procedure to enforce that the model's predictions are invariant to arbitrary rotations and flips of the input, detailed in Supplementary Note 1. Concatenating these correlator estimates results in an $NM$-dimensional feature vector $\mathbf{c} = \{c_\alpha^{(n)}\}$.

In the back portion of a CCNN (see Fig. 1), the feature vector $\mathbf{c}$ is normalized using a BatchNorm layer[32], then used by a logistic classifier which produces the classification output $\hat{y}(\mathbf{c}; \boldsymbol{\beta}, \epsilon) = [1 + \exp(-\boldsymbol{\beta} \cdot \mathbf{c} + \epsilon)]^{-1}$ where $\boldsymbol{\beta} = \{\beta_\alpha^{(n)}\}$ and $\epsilon$ are trainable parameters. If $\hat{y} < 0.5$, the snapshot is classified as $\pi$-flux, and otherwise it is classified as geometric string theory. The $\beta_\alpha^{(n)}$ coefficients are central to the interpretation of the final architecture, as they directly couple the normalized correlator features $c_\alpha^{(n)}$ to the output. For training, we use L1 loss in addition to the standard cross-entropy loss, i.e.,

$$L_{\text{train}}(y, \hat{y}) \equiv -y \log \hat{y} - (1 - y) \log(1 - \hat{y}) + \gamma \sum_{\alpha, k, \mathbf{a}} f_{\alpha, k}(\mathbf{a}), \quad (4)$$

where $y = \{0, 1\}$ is the label of the snapshot, and $\gamma$ is the L1 regularization strength. The role of the L1 loss is to promote sparsity in the filter patterns by turning off pixels which are unnecessary[10,11].

We fix the number of filters $M$ and the maximum order of the nonlinear convolutions $N$, a hyper-parameter specific to CCNN, by systematically observing the training performance. We found that two filters gives sufficient performance while allowing for simple interpretation. Hence we consider two filters, i.e., $M = 2$ in the rest of the paper. For the maximum order of nonlinear convolution $N$ we found the performance to rapidly increase with increase in $N$ up to $N = 4$, past which performance plateaus. Hence we fix $N = 4$ in the rest of the paper. In addition, we limit our investigation to $3 \times 3$ convolutional filters. With the architecture of the CCNN so-fixed we found the performance of this minimalistic model to be comparable with a more complex traditional CNN architecture[3] (see Supplementary Note 1 for these performance results).

After a CCNN is trained, we fix the convolutional filters $f_\alpha$ and move on to a second phase to interpret what it has learned. We first determine which features are the most relevant to the model's performance by constructing and analyzing regularization paths[33] to examine the role of the logistic coefficients $\beta_\alpha^{(n)}$. We apply an L1 regularization loss to these $\beta_\alpha^{(n)}$ and re-train the back portion of the model (see Fig. 1) using a new loss function:

$$L_{\text{path}}(y, \hat{y}) \equiv -y \log \hat{y} - (1 - y) \log(1 - \hat{y}) + \lambda \sum_{\alpha, n} |\beta_\alpha^{(n)}|, \quad (5)$$

where $\lambda$ is the regularization strength. Again, the L1 loss plays a special role in promoting sparsity in the model parameters, but we are now penalizing the use of coefficients $\beta_\alpha^{(n)}$ and hence the corresponding features $c_\alpha^{(n)}$. This results in an optimization trade-off between minimizing the classification loss and attempting to keep $\beta_\alpha^{(n)}$ at zero, where the relative importance of these terms is tuned by $\lambda$. At large $\lambda$, the loss is minimized by keeping all $\beta_\alpha^{(n)}$ at zero, resulting in a 50% classification accuracy due to the model always predicting a single class. As $\lambda$ is slowly ramped down, eventually the "most important" coefficient $\beta_\alpha^{(n)}$ will begin to activate, due to the decrease in classification loss surpassing the increase in the activation loss. As these coefficients couple the correlator features $c_\alpha^{(n)}$ to the prediction output, this process offers clear insight into which features are the most relevant.

We show a typical regularization path analysis in Fig. 3, where the filters $f_\alpha$ of a trained model are shown in the inset. The activation of each coefficient $\beta_\alpha^{(n)}$ is tracked while tuning down the regularization strength $\lambda$ (increasing $1/\lambda$). The resulting

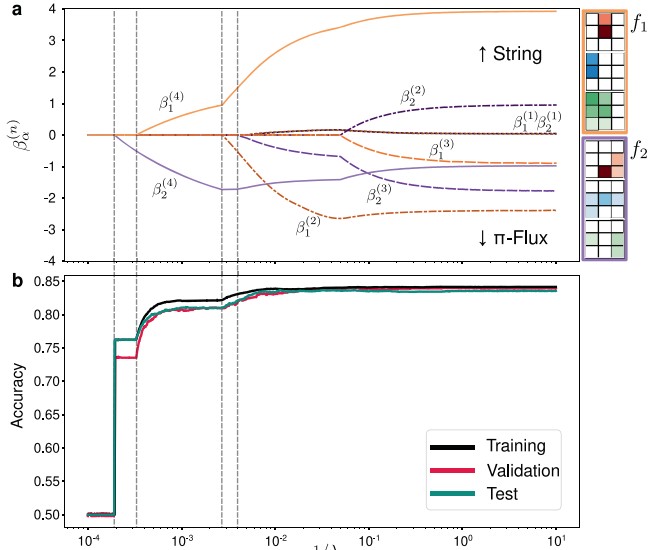

**Fig. 3 Regularization path analysis of a learned fourth-order model. a** The regularization path of $\beta_\alpha^{(n)}$ coefficient values traced out by two learned filters as a function of the inverse regularization strength $1/\lambda$. Positive and negative signs of $\beta_\alpha^{(n)}$ are associated with geometric string and $\pi$-flux labels respectively. **b** The accuracies of the model at each point of the regularization path in (**a**) on both the training dataset, as well a validation dataset unseen by the model during training and a test dataset unseen by us until final evaluation. We use the standard definition of accuracy as the fraction of the snapshots correctly assigned.

trajectories in Fig. 3(a) show that the 4th order correlator features, $c_1^{(4)}$ and $c_2^{(4)}$ are most significant for the CCNN's decision making since $\beta_1^{(4)}$ and $\beta_2^{(4)}$ are the two first coefficients to activate. Furthermore, parallel tracking of the accuracy in Fig. 3(b) shows that the activation of these features results in large jumps in the classification accuracy, comprising almost all of the network's predictive power. While the details of the paths vary between training runs, we find robust dominance of fourth-order correlations as the first features to be activated to give the majority of the network's performance.

The regularization path distinguishing the geometric string and $\pi$-flux ansatzes shown in Fig. 3 is in stark contrast to what happens when the identical architecture is trained to discriminate between a thermally excited antiferromagnetic Heisenberg state and a state with purely random spins (see Supplementary Note 2). In that scenario, the network learns that two-point correlations $c_\alpha^{(2)}$ carry the key information for near-perfect classification. In Supplementary Fig. 5, the regularization path shows only $\beta_1^{(2)}$ activating to achieve full performance, and the learned filter obviously resembles the AFM pattern. Meanwhile, the behavior seen in Fig. 3 evidences that the subtle differences between $\pi$-flux and geometric string theory instead hinges on fourth-order correlations.

Now that we know fourth-order correlations are the important features, we look at which physical correlators are being measured by the features $c_\alpha^{(4)}$ by simply inspecting 4-pixel patterns made from high-intensity pixels from each channel of the learned filters, as we show in Fig. 4. Comparing these patterns with the depiction of the two candidate theories, we can understand why these correlators measured by the two filters are indeed prominent motifs. Specifically, the $2 \times 2$ correlators in the fourth-order feature of the filter associated to the geometric string theory (Fig. 4(a)) are easily recognizable in the "wake" and the termination of a string. These discovered correlations are in agreement

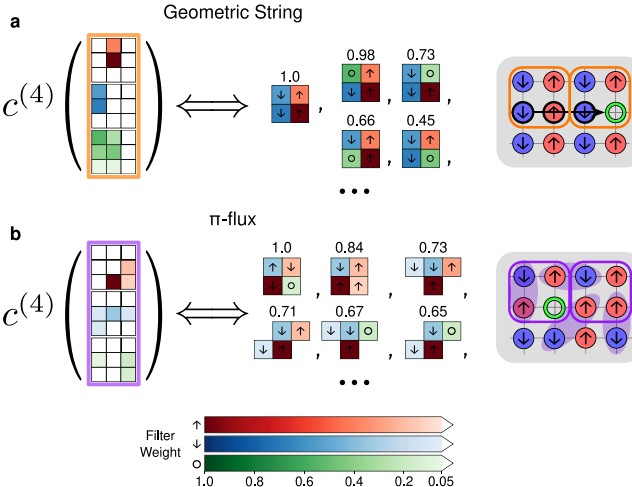

**Fig. 4 Extracting four-point correlators from learned filters. a, b** The highest-weight terms of Eq. (2) when constructing correlator features $c_1^{(4)}, c_2^{(4)}$ from the discovered convolutional filter patterns $f_1, f_2$. We ignore correlators with two or more holes, since these motifs are exceedingly rare in the low-doping regime. Each feature $c_\alpha^{(4)}$ measures a weighted sum of the bare correlators drawn on the right-hand side, obtained by selecting all four-pixel subpatterns from the learned filters. Due to the symmetrization procedure, the model measures all correlations which are symmetry-equivalent under rotations and flips to those drawn. Weights shown above each correlator are obtained as the product of the component filter pixel weights, normalized such that the largest correlator from each filter has weight 1.0.

with those examined in ref. [28], which found pronounced spin anti-correlations induced on the spins located on the diagonal adjacent to a mobile chargon. Meanwhile, the $2 \times 2$ motifs in the filter learned to represent the $\pi$-flux theory (Fig. 4(b)) are either a single spin-flip or a simple placement of a hole into an AFM background. It is evident that this CCNN is learning the finger-print correlations of geometric string theory, recognizing the $\pi$-flux theory instead from fluctuations which are uncharacteristic of the string picture. Furthermore, a subset of learned patterns that are not obvious from the simple cartoons can be used as additional markers to detect the states born out of the two theoretical hypotheses in experiment (see Supplementary Note 4 for more detail).

It is important to note that the above insights relied on the fact that our CCNN's structure can be understood as measuring collections of correlators. Although the regularization path analysis can be applied to any architecture, the typical nonlinear structures of off-the-shelf CNNs inhibit direct connections between the dominant filters and physically meaningful information[34]. In Supplementary Note 5 we present how interpretation of the architecture of ref. [3] can be attempted following similar steps as above. Since the fully connected layer contains tens of thousands of parameters, after training we show that we can reduce this layer to a simple spatial averaging to attempt interpretation, with no loss in performance. The reduced architecture with a single "feature" per convolutional filter, similar to the architecture of ref. [34], is trained, after which we fix the filters for the regularization path analysis. We can clearly determine which filters produce the important features, but it is unclear what these features are actually measuring due to the ReLU nonlinearity. However, without any nonlinearity the architecture only achieves close to 50% performance. This failure to enforce simplicity on traditional architectures shows the importance of designing an architecture, which measures physically meaningful information from the outset.

The ML method presented in this paper considers short-range multi-point correlation functions (up to three lattice sites in both $x$ and $y$ directions), but does not include long-range two-point correlations needed for identifying spontaneous symmetry breaking. Two considerations motivate this choice: (i) Current experiments with the Fermi-Hubbard model are done in the regime where correlations involving charge degrees of freedom are not expected to exceed a few lattice constants due to thermal fluctuations. (ii) The energy of systems with local interactions, such as the Fermi-Hubbard model, is primarily determined by short-range correlations. We note, however, that the current method can be extended to include longer range correlations either by expanding the size of the filters used in Eq. (2), or by using dilated convolutions.

To summarize, we proposed a neural network architecture that discovers most relevant multi-site correlators as key discriminative features of the state of interest. We then applied this architecture to the supervised learning problem of distinguishing two theoretical hypotheses for the doped Hubbard model: $\pi$-flux theory and geometric string theory. Employing a regularization path analysis technique on these trained CCNN architectures, we found that four-site correlators deriving from the learned filters hold the key fingerprints of geometric string theory. A subset of these four-site motifs fit into what is expected from the wake of a propagating hole in an anti-ferromagnetic background. The remaining four-site motifs which go beyond our existing intuition can be used as additional signatures of the two quantum states. As higher-order correlators are beginning to be probed in QGM experiments[19], our work demonstrates an automated method for learning high signal-to-noise correlators useful for theory hypothesis testing. It will be interesting to extend our analysis to a broader collection of candidate theories, as well as snapshots generated using recently developed finite-$T$ tensor network methods[35,36] and spin-resolved experimental data.

## Discussion

The broad implications of CCNN-based ML for analysis and acquisition of image-like data are threefold. Firstly, CCNN is the first neural network architecture that was explicitly designed for image-like quantum matter data. Our results showcase how a successful design of ML architecture that is designed with scientific objectives at the forefront can offer new scientific insight.

Secondly, our approach can guide quantum simulator design by revealing necessary discriminating features. In particular, we found that experimental uncertainties on the actual doping level in QGM without spin-resolution led the CCNN to focus on the doping level rather than a meaningful hypothesis testing (see Supplementary Note 6). Hence, access to either spin or charge-resolved snapshots, which are just now becoming available[27,28,37,38], will be essential. Finally, our results showcase how a targeted "tomography" can be achieved to extract new insights from near-term quantum systems from quantum-classical hybrid approaches. Full reconstruction of the density matrices from projective measurements is an exponentially difficult task. However, available and near-term quantum systems are showing great promise as quantum simulators with their design and objectives guided by classical simulations. For such quantum systems, CCNN-based hypothesis testing can offer much needed state characterization in a scalable fashion.

## Disclaimer

This report was prepared as an account of work sponsored by an agency of the United States Government. Neither the United

States Government nor any agency thereof, nor any of their employees, makes any warranty, express or implied, or assumes any legal liability or responsibility for the accuracy, completeness, or usefulness of any information, apparatus, product, or process disclosed, or represents that its use would not infringe privately owned rights. Reference herein to any specific commercial product, process, or service by trade name, trademark, manufacturer, or otherwise does not necessarily constitute or imply its endorsement, recommendation, or favoring by the United States Government or any agency thereof. The views and opinions of authors expressed herein do not necessarily state or reflect those of the United States Government or any agency thereof.

## Data availability
All simulated snapshots examined in this work are available publically at Zenodo, ref. [39]. All experimental snapshots examined in the Supplementary Notes are available from ref. [20].

## Code availability
All code used for training and analysis of the CCNNs is available at Github, ref. [40].

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

## Acknowledgements
We thank Fabian Grusdt and Andrew Gordon Wilson for insightful discussions during the completion of this work. C.M. acknowledges that this material is based upon work supported by the U.S. Department of Energy, Office of Science, Office of Advanced Scientific Computing Research, Department of Energy Computational Science Graduate Fellowship under Award Number DE-SC0020347. A.B., R.W., K.W., E.D., E-A.K. acknowledge support by the National Science Foundation through grant No. OAC-1934714. A.B. acknowledges funding by Germany's Excellence Strategy - EXC-2111 - 390814868.

## Author contributions
C.M. conceived the CCNN architecture, wrote the ML and analysis code, and performed the training experiments and analysis. R.W. and K.Q.W conceived the application of regularization paths, and provided guidance of the ML procedure. A.B. produced the simulated snapshot data. C.C., M.X., G.J., and M.G. produced the experimental data and provided feedback on connections to experiments. C.M., A.B., R.W., K.Q.W., E.D., and E-A.K. initiated the project concept, and guided the work. C.M. and E.A.K. wrote the paper, with input and modifications from all authors. E.A.K. led the project.

## Competing interests
The authors declare no competing interests.
