## [Peer Review File · Nature Communications]

REVIEWER COMMENTS

Reviewer #1 (Remarks to the Author):

Report on the manuscript NCOMMS-21-00265-T:

Recent experimental progress in quantum gas microscope (QGM) allows microscopic measurements of Fermi-Hubbard system, raising great research interest in analyzing the microscopic data. In particular, various machine learning techniques have been employed to analyse the image-like data from quantum simulations. Here the authors analyzed the snapshots from the geometric string and π -flux RVB theories, and uncover that the fourth-order spin-charge correlators constitute the key quantities in future studies of the doped Fermi-Hubbard model. This work is built on solid analysis and advanced machine learning model design --- the so-called correlator convolution neural network (CCNN) --- that can be used to investigate the higher-order correlators and is thus surely of interest to the community.

However, when reviewing this work in details, the following concerns came across my mind, which makes me difficult to recommend the publication of it at the present stage.

(1) The CCNN is trained using snapshots from two candidate theories proposed to capture the essence of Fermi-Hubbard quantum states at finite doping. It is, therefore, highly desirable to check if the CCNN can discriminate the two theories (and possibly others) from the real experimental data of Fermi-Hubbard quantum simulations. In the present manuscript, despite some analysis of experimental data in Sec. S.VII, the authors did not have a chance to show the powerfulness of their CCNN in realistic problems. The authors explained in the main text that this is due to the lack of spin-resolved QGM data. However, such data, as a matter of fact, are available in current QGM experiments

[cf. Phys. Rev. Lett. 125, 010403 (2020) and Phys. Rev. Lett. 125, 113601 (2020)].

(2) If the spin-resolved experimental data are not available for the authors, did they try applying their CCNN to numerical data of Fermi-Hubbard model and see which one of the two theories better describes the Fermi-Hubbard model by identifying, e.g., the key filters? As stated in the abstract, the authors believe their work ``paves the way for new physical insights from machine learning studies

of experimental as well as numerical data", can the authors show some evidence, either with experimental or numerical data, to make this point more convincing?

(3) In the follow-up work arXiv:2101.00721 with some authors in overlap with the present manuscript, they considered higher-order correlations in the Fermi-Hubbard model under doping, and have computed third- and fifth-order correlation, but the fourth-order correlator --- central finding in this work-----was skipped. Can the authors comment on that?

(4) Moreover, higher-order correlators have been previously measured in QGM experiments in arXiv:2009.04440, where fourth-order spin-charge correlators have been discussed. Can the authors compare the findings in the present manuscript with theirs? And can the authors provide an analytical form of fourth-order correlators that can be checked numerically?

Besides the major concerns above, I have also a few minor questions.

(i) The authors mentioned in the main text and provide more details on the discrimination of AF Heisenberg and random spin states in the Supplementary Materials, which is good to see. However, they did not mention how their AF Heisenberg snapshots are generated (in Fig. S4 left). Are they from numerical simulations (e.g., quantum Monte Carlo) or quantum simulators (like QGM)?

(ii) Another technical question is on the symmetry of 2×2 motifs in the learned filters. For example, in the key filter in Fig. 4(a) learned from the geometric string theory, we see spin-antiparallel columns perpendicular to the string path. It is supposed to have an equal weight when we flip the spin-up(-down) column to spin down(up) in the correlator of weight 1.0 shown in Fig. 4(a).

Same for the RVB pattern in Fig. 4(b) when flip the spins of two sites connected by a valence bond. Did the authors observe such symmetry in the learned filter? Or such symmetry is spontaneously broken in the convolution process?

(iii) In Fig.~3, indeed $\beta_1^{(4)}$ and $\beta_2^{(4)}$ are first activated when decreasing the regularization strength λ . However, when one focuses on the high-accuracy plateau (i.e. $\lambda \gtrsim 10^2$) as indicated in Fig.~3(b), the components $\beta_1^{(2)}$ and $\beta_2^{(3)}$ are also significant. Does it also suggest that the 2nd- and 3rd-order correlations can be employed to characterise the π -flux states. Can the authors comment?

Reviewer #2 (Remarks to the Author):

Report for: "Correlator Convolutional Neural Networks: An Interpretable Architecture for Image-like Quantum Matter Data"

by Cole Miles, Annabelle Bohrdt, Ruihan Wu, Christie Chiu, Muqing Xu,² Geoffrey Ji, Markus Greiner, Kilian Q. Weinberger, Eugene Demler, and Eun-Ah Kim

The authors implement a novel convolutional neural network for classifying snapshots of ultracold gases in optical lattices, which are described by the celebrated 2D Hubbard model.

In this work, the authors employ (essentially exclusively) synthetic snapshots generated by Monte Carlo sampling from two competitive approximate theories which are believed to describe the Hubbard model at finite doping, namely, the geometric string theory and the π -flux theory. The network is designed to allow identifying which correlations are relevant to classify snapshots. This is achieved by removing conventional layers with non-linear activations functions, and including high-order convolutions truncated at the desired order. Furthermore, the authors show how to use regularization path analysis to identify the most relevant features. The trained model classifies unseen snapshots with high accuracy, showing that fourth-order spin-charge correlations are the key feature to discern snapshots from the two theories.

Several research groups are currently investigating the use of machine learning algorithms to analyze/classify experimental measurements, in particular in the context of cold-atom experiments; see, e.g., Refs.[3,8,9], and also arXiv:2002.07055.

Also, there is great interest in the implementation of physically interpretable machine-learning models; see, e.g., [13, 30, 31]. The present manuscript reports sound and useful technical advancements on the use of artificial neural networks for this problem. In particular, it shows how a simple and interpretable model can compete with much deeper networks with many parameters, and it introduces in the physics context the use of regularization path analysis.

For these reasons, I think that the present manuscript can be considered for publication in Nature Communication.

Below I report a few issues which should be addressed before publication.

Comments/issues:

-) By reading the introduction, the reader is lead to believe that the manuscript mostly focuses on experimental data. In page 3, the authors state that they "use primarily simulated snapshot". In fact, essentially the whole manuscript focuses on simulated data, with some experimental data analyzed only in the supplemental material. I suggest the authors to better explain, in the introduction, that their goal here is to analyze simulated data. Otherwise, they should expand the discussion on

experimental data, and possibly provide the physics insight they gained on the QGM experiment on the 2D Hubbard model.

-) The discussion of why the π -flux and the geometric string theories are considered might be expanded, including the physical implications of the two theories.

-) The description of how training/validation data are generated is provided only in the supplemental material. I think that a short description should be included in the main text, too. This would help the reader understanding the procedure the authors aim to describe.

-) The definition of accuracy is not provided. I assume it is the standard one for binary classification tasks: $A=(TP+TN)/(TP+TN+FP+FN)$.

-) While the π -flux and the geometric string theory are different, I suspect that that they might allow generating identical snapshots in finite systems. The authors did not discuss if and how they inspect if identical snapshots can be (or are) present in the two sets from the two theories, and how this could affect the training process and the maximum achievable accuracy.

-) Neural networks have been used to analyze cold-atom setups also in arXiv:2002.07055 and in Phys. Rev. Applied 14, 014011 (2020) (using experimental data) and in Scientific Reports 9, 1-12 (2019) (using simulated data).

-) In machine-learning jargon, the term "validation set" is typically used of a dataset which is not used to minimize the loss function, but might be used for hyper-parameter selection or to control/halt the training process. The generalization accuracy should be quantified using a distinct "test set". I suspect that what the authors mean is, actually, "test set".

Reviewer #3 (Remarks to the Author):

Dear editor,

in the manuscript "Correlator Convolutional Neural Networks: An Interpretable Architecture for Image-like Quantum Matter Data" by Miles et al., the authors present correlator-based neural network and their use for analysis of quantum gas microscope data. Specifically, the presented method uses the explicit relation of the network's filters to the relevant correlations in the system. The authors use this relation to determine the better of the two candidate theories (geometric strings, pi-flux) to explain specific set of quantum gas microscope snapshots.

The work by Miles et al. presents a valuable addition to the interpretable machine learning models for physics. While neural networks have shown unprecedented flexibility for various tasks in many fields, in the standard setting their weights are not necessarily physically motivated and hence their interpretation presents a challenge. Here, the authors present novel and concrete way to overcome this difficulty in a relevant experimental setting.

The manuscript is very well written and all the key points are easy to follow. I am happy to recommend it for the publication in Nat. Comm.

I have two questions, the clarification of which would add, in my opinion to my understanding of the robustness and applicability of the method:

(1) The authors use L1 loss for regularisation, but this choice is not motivated in the manuscript. Is there a specific motivation for regularising like this? If I were applying the method in another setting might I need more regularisation terms? The detailed analysis of the regularisation path for different regularisation terms is likely outside the scope of the present work, but I think it would be useful to understand how the present L1 term was chosen.

(2) How big is the training set used? I imagine the measurements taken are not informationally complete, but the network at least needs to 'see' sufficient amount of data to be able to distinguish the underlying correlations.

LETTER TO THE REFEREES

We are grateful to the referees for their valuable comments which have led us to improve the clarity of several key points of our work.

We are happy to see that all of the reviewers find our work to be of interest to the general community, and to be built upon a solid foundation. In particular, referees #2 and #3 are especially positive that our work has made significant advances beyond the existing ML techniques applied to quantum data. Referee #1's comments and suggestions have pushed us to communicate our results and the context of our work within the greater community more clearly, and we believe this has resulted in a much improved manuscript.

In response to the referees suggestions and comments we have made several changes to the main text and supplement, which are marked in red. We respond in detail to the referees' comments below. In summary, our changes are:

1. The addition of a newly-generated test dataset for performance measurements, added to Fig. 3.
2. Additional details in the supplement on the regularization path technique, dataset generation, consequences of identical snapshots, and further measurements of more correlators and their connected components.
3. Expanded discussion of the relationship of our work to a recent related experimental work (arXiv:2101.09280), and general applicability of our approach.

We are confident that the revised version is significantly clearer in its presentation due to the valuable suggestions made by the referees.

Regards,

Cole Miles, Annabelle Bohrdt, Ruihan Wu, Christie Chiu, Muqing Xu, Geoffrey Ji, Markus Greiner, Kilian Q. Weinberger, Eugene Demler, and Eun-Ah Kim

Report of First Referee

Recent experimental progress in quantum gas microscope (QGM) allows microscopic measurements of Fermi-Hubbard system, raising great research interest in analyzing the microscopic data. In particular, various machine learning techniques have been employed to analyse the image-like data from quantum simulations. Here the authors analyzed the snapshots from the geometric string and π -flux RVB theories, and uncover that the fourth-order spin-charge correlators constitute the key quantities in future studies of the doped Fermi-Hubbard model. This work is built on solid analysis and advanced machine learning model design — the so-called correlator convolution neural network (CCNN) — that can be used to investigate the higher-order correlators and is thus surely of interest to the community.

However, when reviewing this work in details, the following concerns came across my mind, which makes me difficult to recommend the publication of it at the present stage.

(1) The CCNN is trained using snapshots from two candidate theories proposed to capture the essence of Fermi-Hubbard quantum states at finite doping. It is, therefore, highly desirable to check if the CCNN can discriminate the two theories (and possibly others) from the real experimental data of Fermi-Hubbard quantum simulations. In the present manuscript, despite some analysis of experimental data in Sec. S.VII, the authors did not have a chance to show the powerfulness of their CCNN in realistic problems. The authors explained in the main text that this is due to the lack of spin-resolved QGM data. However, such data, as a matter of fact, are available in current QGM experiments [cf. Phys. Rev. Lett. 125, 010403 (2020) and Phys. Rev. Lett. 125, 113601 (2020)].

Indeed, spin-resolved experimental data is beginning to become available to certain groups, and we are excited to apply our technique to uncover interesting correlations in this data as originally mentioned near the end of our main text.

However, we currently have two barriers to doing so. Firstly, the cited works either have radially-varying doping levels within each snapshot (Koepsell et. al.) or are at too high of a temperature (Hartke et. al.), either of which make it significantly more difficult to perform one-to-one comparisons to the low-temperature theoretical snapshots we are interested in. We thank the referee for pointing out these relevant works, and have added citations to these near the end of the text where we discuss our desire for fully resolved data.

A more significant blockade is that unfortunately, none of these groups have made their snapshot repositories publicly available. So, we do not have access to examine these at the moment.

(2) If the spin-resolved experimental data are not available for the authors, did they try applying their CCNN to numerical data of Fermi-Hubbard model and see which one of the two theories better describes the Fermi-Hubbard model by identifying, e.g., the key filters? As stated in the abstract, the authors believe their work “paves the way for new physical insights from machine learning studies of experimental as well as numerical data”, can the authors show some evidence, either with experimental or numerical data, to make this point more convincing?

While we would very much like to study “true” finite-temperature doped Fermi-Hubbard snapshots, generating these at the sizes we are studying is a very non-trivial task. The main technique one would envision using is Quantum Monte Carlo, however the doped Hubbard model has a sign problem making accurate sampling extremely difficult.

[redacted]

(3) In the follow-up work arXiv:2101.00721 with some authors in overlap with the present manuscript, they considered higher-order correlations in the Fermi-Hubbard model under doping, and have computed third- and fifth-order correlation, but the fourth-order correlator — central finding in this work — was skipped. Can the authors comment on that?

These works are unrelated, parallel lines of research. The cited work (arXiv:2101.00721) is a follow-up work to papers (Phys. Rev. Lett. 126, 026401) and (Phys. Rev. B 99, 224422). Additionally, the cited work is studying spin correlations in the vicinity of a single hole in the presence of a pinning potential, not the finite doping regime of the Fermi-Hubbard model, which we study here. Moreover, the cited work (arXiv:2101.00721) does not aim to distinguish two different theories.

Nonetheless, the correlations uncovered by our network as characterizing the string theory are painting a similar picture as those probed in those works. Both analyses point to holes as introducing high-order correlations resulting in spins surrounding holes tending to flip relative to the overall AFM order. Several different correlators can be measured to probe this same phenomena.

(4) Moreover, higher-order correlators have been previously measured in QGM experiments in arXiv:2009.04440, where fourth-order spin-charge correlators have been discussed. Can the authors compare the findings in the present manuscript with theirs?

The correlators studied in the cited work and the fourth-order correlators the CCNN learned to focus on are similar and different in various ways. Most importantly, in the extreme underdoped regime of interest, two holes rarely came near each other in the snapshots we studied. CCNN is set up to learn the most distinct, data-efficient features and it found the fourth order correlators we reported. Additionally, the CCNN only “sees” bare correlations, which differ to the connected components studied in the cited work.

We have added a reference to arXiv:2009.04440 and brief discussion to the end of the main text, along with modifying the caption of Fig. 4(a) to explicitly point out we are omitting two-hole correlators. We thank the reviewer for pushing us to take a closer look at our results, and believe that these extended discussions and measurements have greatly improved the quality of our work.

And can the authors provide an analytical form of fourth-order correlators that can be checked numerically

We can provide analytic forms of the correlators measured by the network. For instance, by specializing Eq. 2 to the filter shown in Fig. 4(a), we can write that the network *in expectation* measures:

$$c_1^{(4)} = \frac{1.0}{0.98} \langle \hat{n}_{(i,j),\downarrow} \hat{n}_{(i,j+1),\downarrow} \hat{n}_{(i+1,j),\uparrow} \hat{n}_{(i+1,j+1),\uparrow} \rangle + \langle \hat{n}_{(i,j),\downarrow} (1 - \hat{n}_{(i,j+1,\uparrow)} - \hat{n}_{(i,j+1,\downarrow)}) \hat{n}_{(i+1,j),\uparrow} \hat{n}_{(i+1,j+1),\uparrow} \rangle + \dots$$

under the assumption that we are in the singly-occupied portion of Hilbert space. Histogramming these various terms for the two theories is exactly what is done in the Supplement, Sec. V, which verifies that the networks decisions are in line with the true distributions.

Besides the major concerns above, I have also a few minor questions.

(i) The authors mentioned in the main text and provide more details on the discrimination of AF Heisenberg and random spin states in the Supplementary Materials, which is good to see. However, they did not mention how their AF Heisenberg snapshots are generated (in Fig. S4 left). Are they from numerical simulations (e.g., quantum Monte Carlo) or quantum simulators (like QGM)?

[Redacted]

We thank the reviewer for pointing out that we were unclear on this. The snapshots are generated from Quantum Monte Carlo simulations, and we have added additional text to clarify this to the supplement, Sec. S.II.

(ii) Another technical question is on the symmetry of 2×2 motifs in the learned filters. For example, in the key filter in Fig. 4(a) learned from the geometric string theory, we see spin-antiparallel columns perpendicular to the string path. It is supposed to have an equal weight when we flip the spin-up(-down) column to spin down(up) in the correlator of weight 1.0 shown in Fig. 4(a). Same for the RVB pattern in Fig. 4(b) when flip the spins of two sites connected by a valence bond. Did the authors observe such symmetry in the learned filter? Or such symmetry is spontaneously broken in the convolution process?

Without any symmetrization, these symmetries would be broken by the convolution. However, we encode our prior knowledge of the symmetries by explicitly symmetrizing our model with respect to arbitrary rotations / flips of each learned pattern, as detailed in the Supplement, Sec I. For every pattern showed in Fig. 4, the CCNN is actually measuring all symmetry-equivalent versions of these patterns. (However, we did not implement the spin-flip symmetry in our model, though one could envision doing so.)

We thank the reviewer for pointing out that this was not explained in the manuscript – we have added additional details to the main text (top of page 4, and Fig. 4 caption) pointing out the symmetrization to make this point more obvious.

(iii) In Fig. 3, indeed $\theta_1^{(4)}$ and $\theta_2^{(4)}$ are first activated when decreasing the regularization strength λ . However, when one focuses on the high-accuracy plateau (i.e. $\lambda \sim 10^2$) as indicated in Fig. 3(b), the components $\theta_1^{(2)}$ and $\theta_2^{(3)}$ are also significant. Does it also suggest that the 2nd- and 3rd-order correlations can be employed to characterise the π -flux states. Can the authors comment?

The reviewer is correct in pointing out that our analysis shows that $c_1^{(2)}$ and $c_2^{(3)}$ can be relevant features for characterizing these states. The main point in our original text is simply that $c_1^{(4)}$ and $c_2^{(4)}$ provide the majority of the characterization power, while the other orders are smaller corrections on top of these.

Report of the Second Referee

The authors implement a novel convolutional neural network for classifying snapshots of ultracold gases in optical lattices, which are described by the celebrated 2D Hubbard model. In this work, the authors employ (essentially exclusively) synthetic snapshots generated by Monte Carlo sampling from two competitive approximate theories which are believed to describe the Hubbard model at finite doping, namely, the geometric string theory and the π -flux theory. The network is designed to allow identifying which correlations are relevant to classify snapshots. This is achieved by removing conventional layers with non-linear activations functions, and including high-order convolutions truncated at the desired order. Furthermore, the authors show how to use regularization path analysis to identify the most relevant features. The trained model classifies unseen snapshots with high accuracy, showing that fourth-order spin-charge correlations are the key feature to discern snapshots from the two theories.

Several research groups are currently investigating the use of machine learning algorithms to analyze/classify experimental measurements, in particular in the context of cold-atom experiments; see, e.g., Refs.[3,8,9], and also arXiv:2002.07055. Also, there is great interest in the implementation of physically interpretable machine-learning models; see, e.g., [13, 30, 31]. The present manuscript reports sound and useful technical advancements on the use of artificial neural networks for this problem. In particular, it shows how a simple and interpretable model can compete with much deeper networks with many parameters, and it introduces in the physics context the use of regularization path analysis. For these reasons, I think that the present manuscript can be considered for publication in Nature Communication. Below I report a few issues which should be addressed before publication.

Comments/issues:

-) By reading the introduction, the reader is lead to believe that the manuscript mostly focuses on experimental data. In page 3, the authors state that they "use primarily simulated snapshot". In fact, essentially the whole manuscript focuses on simulated data, with some experimental data analyzed only in the supplemental material. I suggest the authors to better explain, in the introduction, that their goal here is to analyze simulated data. Otherwise, they should expand the discussion on experimental data, and possibly provide the physics insight they gained on the QGM experiment on the 2D Hubbard model.

The reviewer is correct in pointing out that we put an overly-large emphasis on experimental data in the introduction. We have adjusted this introductory text to focus more on the simulated data that takes up the bulk of the work.

-) The discussion of why the π -flux and the geometric string theories are considered might be expanded, including the physical implications of the two theories.

The two theories considered here are both good candidate theories to describe the under doped ($\approx 10\%$) Fermi-Hubbard model at finite temperatures, as considered here. Recent studies have shown that the geometric string theory provides a good – even quantitative – semi-analytical explanation of many numerical and experimental findings in the study of a single hole (Grusdt et al. Phys. Rev. B 99, 224422 (2019), Koepsell et al. Nature 572, 358 (2019), Bohrdt et al. New J. Phys. 22 123023 (2020), Bohrdt et al. Phys. Rev. B 102, 035139 (2020), Ji et al. arXiv:2006.06672, Bohrdt et al. arXiv:2101.09280). The choice of an RVB state goes back to the original proposal by Anderson in 1987

(Anderson, Science Vol. 235, Issue 4793 (1987)) to describe the Fermi-Hubbard model and the cuprate materials. We agree with the referee that a different RVB state, for example a uniform RVB state, could have been chosen instead. The choice for the π -flux state was guided by the fact that the ground state of the Heisenberg model in the large-N limit corresponds to the fermionic π -flux, or d-wave state (Marston and Affleck, Phys.Rev. B, 39:11538–11558 (1989)). Previous work has shown that both theories chosen here describe conventional observables, such as two-point spin correlation functions and full counting statistics of the staggered magnetization, as well as more unconventional observables, such as string patterns, as observed in experiments extremely well (Chiu2019). For the present application we moreover need to consider theories from which Fock space snapshots can be sampled, which is unfortunately not straightforwardly possible in many cases, such as for example for field theoretical approaches. We agree with the referee that an interesting line of future work would be to compare more candidate theories for the doped Fermi-Hubbard model and systematically investigate the correlations that are most distinct between them, and we have added a remark along this line to the final discussion in the main text.

-) The description of how training/validation data are generated is provided only in the supplemental material. I think that a short description should be included in the main text, too. This would help the reader understanding the procedure the authors aim to describe.

We agree that a short description would help readers understand the full pipeline of our work, and so we have written a brief summary of the generation process in the main text, in paragraph 6 (on page 3).

-) The definition of accuracy is not provided. I assume it is the standard one for binary classification tasks: $A=(TP+TN)/(TP+TN+FP+FN)$.

We do use the equation stated by the reviewer as accuracy, and have clarified this in caption of Fig. 3.

-) While the π -flux and the geometric string theory are different, I suspect that that they might allow generating identical snapshots in finite systems. The authors did not discuss if and how they inspect if identical snapshots can be (or are) present in the two sets from the two theories, and how this could affect the training process and the maximum achievable accuracy.

This is an interesting point to make. For our dataset, we have checked and found that there are no instances of identical snapshots between the two datasets. However, it is certainly the case that it is possible to occur, though we believe this should not affect our analysis. In the case that the two distributions to classify between have overlapping support, the optimal model should output $\hat{y}(\mathbf{x}) = p_{\text{string}(\mathbf{x})}/(p_{\text{string}(\mathbf{x})} + p_{\pi\text{-flux}(\mathbf{x})})$, where p 's are the probabilities to sample \mathbf{x} from each model. This does imply a maximum achievable accuracy which depends on how much the distributions overlap, but this maximum accuracy is unknown to us and would be extremely difficult to calculate.

We have added a paragraph to Sec. S.I of the supplement discussing this, and thank the reviewer for leading us to think more deeply on this.

-) Neural networks have been used to analyze cold-atom setups also in arXiv:2002.07055 and in Phys. Rev. Applied 14, 014011 (2020) (using experimental data) and in Scientific Reports 9, 1-12 (2019) (using simulated data).

We thank the reviewer for pointing out these relevant works. We have added citations of these works in the introductory paragraphs describing previous ML applications to quantum matter data.

-) In machine-learning jargon, the term "validation set" is typically used of a dataset which is not used to minimize the loss function, but might be used for hyper-parameter selection or to control/halt the training process. The generalization accuracy should be quantified using a distinct "test set". I suspect that what the authors mean is, actually, "test set".

The reviewer is making a good point about the distinction between validation and test data sets. For a maximally unbiased judgement of generalization error, there should exist a "test" data set that is never looked at until the final model is trained. The reason for this is that in many ML contexts, extensive hyperparameter tuning is done to minimize the train/val error gap. The test set serves to judge how much this hyperparameter tuning affected "true" generalization error on data unseen by either the model or the researcher.

Due to the extremely small number of parameters in our model, this generalization gap was always small for all trained models, hence no "hyperparameter tuning" was necessary in our case. Because of this, we originally thought that a third test set was unnecessary. However, to completely confirm this, we have generated a brand new set of data to call a test set, and evaluated performance using the exact same trained model as examined before. We have added the performance on this set as a new curve in Fig. 3(b). We thank the reviewer for leading us to improve our analysis to confirm we are completely unbiased in our measurements.

Report of the Third Referee

In the manuscript "Correlator Convolutional Neural Networks: An Interpretable Architecture for Image-like Quantum Matter Data" by Miles et al., the authors present correlator-based neural network and their use for analysis of quantum gas microscope data. Specifically, the presented method uses the explicit relation of the network's filters to the relevant correlations in the system. The authors use this relation to determine the better of the two candidate theories (geometric strings, pi-flux) to explain specific set of quantum gas microscope snapshots.

The work by Miles et al. presents a valuable addition to the interpretable machine learning models for physics.

While neural networks have shown unprecedented flexibility for various tasks in many fields, in the standard setting their weights are not necessarily physically motivated and hence their interpretation presents a challenge. Here, the authors presents novel and concrete way to overcome this difficulty in a relevant experimental setting.

The manuscript is very well written and all the key points are easy to follow. I am happy to recommend it for the publication in Nat. Comm.

I have two questions, the clarification of which would add, in my opinion to my understanding of the robustness and applicability of the method:

(1) The authors use L1 loss for regularisation, but this choice is not motivated in the manuscript. Is there a specific motivation for regularising like this? If I were applying the method in another setting might I need more regularisation terms? The detailed analysis of the regularisation path for different regularisation terms is likely outside the scope of the present work, but I think it would be useful to understand how the present L1 term was chosen.

Our choice of L1 regularization is motivated by its use in the statistics community (where it is often known as LASSO) for performing feature selection. Unregularized statistical models tend to drive learned coefficients to unreasonably large values. While all regularization terms attempt to control this, the L1 loss is special in that it tends to drive unimportant coefficients *completely to zero* rather than just an overall downwards scaling. We broadly expect that L1 regularization will work for final feature selection, though it is not the “only” choice.

We have edited the manuscript to emphasize the important role of the L1 loss. Additionally, we have added an extended discussion to the supplement, Sec. S.I, including citations to alternate path algorithms that we envision could be useful. We thank the reviewer for pushing us to explain this point more clearly.

(2) How big is the training set used? I imagine the measurements taken are not informationally complete, but the network at least needs to ‘see’ sufficient amount of data to be able to distinguish the underlying correlations.

The reviewer is correct that the measurements are not informationally complete, and that the model’s performance will absolutely depend on the amount of training data is able to learn the correlations from.

In our work, our dataset consisted of 24, 075 snapshots, about 18k of which were placed in the training dataset with the rest used as either validation or test sets. We have updated Sec. S.I of the supplement to include more details on these sets and data generation in general, and we have also added an extra pointer to this section in the main text.

REVIEWERS' COMMENTS

Reviewer #1 (Remarks to the Author):

I am happy to see that the authors have addressed all the points -- raised by myself and other referees -- satisfactorily. In particular, they provide evidence in the reply that CCNN can be applied to realistic experimental data. With detailed response in the reply and careful revisions in the Manuscript/Supplementary Information, I am convinced the interpretable CCNN model invented can be very useful in quantum matter studies, and would like to recommend its publication in Nature Communications.

There is only one minor point I would further mention. In the reply the authors said "While we would very much like to study "true" finite-temperature doped Fermi-Hubbard snapshots, [...] The main technique one would envision using is Quantum Monte Carlo, however the doped Hubbard model has a sign problem making accurate sampling extremely difficult". While this could be true for Monte Carlo calculations, recent progress in tensor networks makes the efficient simulations of the doped Hubbard model at finite T possible (see, e.g., PRB 103, L041107 (2021) and arXiv:2009.10736). The CCNN architecture may also do some help in analyzing snapshots generated from these tensor network simulations.

Reviewer #2 (Remarks to the Author):

The authors have exhaustively addressed the comments I raised in my previous report. I can confirm my recommendation for publication in Nature Communications.

Reviewer #3 (Remarks to the Author):

Dear editor,

the authors thoroughly addressed my comments - the added explanation clarifies all questions I had. As stated in my previous report, the present manuscript is relevant addition to interpretable

machine learning on experimentally accessible data. I am happy with the current version of the manuscript and recommend it for publication.

LETTER TO THE REFEREES #2

We again wish to thank the referees for their efforts in reviewing our work. We are happy to hear that we have updated the manuscript satisfactorily, and are elated that our manuscript has been accepted for publication conditioned on a suitable revision.

We reproduce the referees' comments below, and provide our responses.

Regards,

Cole Miles, Annabelle Bohrdt, Ruihan Wu, Christie Chiu, Muqing Xu, Geoffrey Ji, Markus Greiner, Kilian Q. Weinberger, Eugene Demler, and Eun-Ah Kim

Report of First Referee

I am happy to see that the authors have addressed all the points – raised by myself and other referees – satisfactorily. In particular, they provide evidence in the reply that CCNN can be applied to realistic experimental data. With detailed response in the reply and careful revisions in the Manuscript/Supplementary Information, I am convinced the interpretable CCNN model invented can be very useful in quantum matter studies, and would like to recommend its publication in Nature Communications.

There is only one minor point I would further mention. In the reply the authors said "While we would very much like to study "true" finite-temperature doped Fermi-Hubbard snapshots, [...] The main technique one would envision using is Quantum Monte Carlo, however the doped Hubbard model has a sign problem making accurate sampling extremely difficult". While this could be true for Monte Carlo calculations, recent progress in tensor networks makes the efficient simulations of the doped Hubbard model at finite T possible (see, e.g., PRB 103, L041107 (2021) and arXiv:2009.10736). The CCNN architecture may also do some help in analyzing snapshots generated from these tensor network simulations.

We thank the referee for pointing out an alternate viable avenue for generating finite- T Hubbard snapshots that we had not thought of in our original response. We agree that it would be interesting to apply a similar CCNN analysis to snapshots output from similar tensor network simulations. To reflect this, we have updated our discussion near the end of the manuscript to mention this as a potential future direction of work, along with the relevant citations.

Report of Second Referee

The authors have exhaustively addressed the comments I raised in my previous report. I can confirm my recommendation for publication in Nature Communications.

Report of Third Referee

The authors thoroughly addressed my comments - the added explanation clarifies all questions I had. As stated in my previous report, the present manuscript is relevant addition to interpretable machine learning on experimentally accessible data. I am happy with the current version of the manuscript and recommend it for publication.

We again thank the two above referees for their previous comments and recommendations, and are happy that we have successfully updated the manuscript to be clearer in its communication.